# “Good Guys Don’t Rape”: Greek and Non-Greek College Student Perpetrator Rape Myths

**DOI:** 10.3390/bs8070060

**Published:** 2018-06-27

**Authors:** Taylor Martinez, Jacquelyn D. Wiersma-Mosley, Kristen N. Jozkowski, Jennifer Becnel

**Affiliations:** 1School of Human Environmental Sciences, University of Arkansas, Fayetteville, AR 72701, USA; taymartinez19@gmail.com (T.M.); becnel@uark.edu (J.B.); 2The Department of Health, Human Performance, and Recreation, University of Arkansas, Fayetteville, AR 72701, USA; kjozkows@uark.edu; 3The Kinsey Institute for Research in Sex, Gender, and Reproduction, Indiana University, Bloomington, IN 47405, USA

**Keywords:** perpetrator rape myths, fraternity men, sorority women, college campus

## Abstract

The current study examined sexual assault perpetrator rape myths among college students, and in particular Greek students. Fraternity men are overrepresented among sexual assault perpetrators, while sorority women are at increased risk for victimization of sexual assault. The current study examined Greek-affiliated and non-Greek-affiliated perceptions of perpetrator rape myths among 892 college students; 58% of the sample was Greek-affiliated. Men and Greek-affiliated students reported higher agreement on stereotypes than women and non-Greek-affiliated students regarding perpetrator rape myths. Specifically, fraternity men reported higher stereotypical perceptions compared to all women and non-affiliated men, while there was no difference between sorority and non-affiliated women.

## 1. Introduction

Research on campus sexual violence has increasingly emphasized the role of rape myths regarding victims and women, but there is considerably less research focused on rape myths regarding perpetrators and men. Rape myths are the stereotypical beliefs regarding sexual assault, victims of sexual assault, as well as the situational variables that distinguish sexual assault from consensual sex [1,2]. One group that has shown higher endorsement of rape myths are college students participating in Greek life. In addition, fraternity men and sorority women are overrepresented in perpetration and victimization in sexual assault cases [3,4,5,6,7,8,9,10]. Thus, it may be important to study the perceptions of perpetrators of sexual assault among college-aged Greek-affiliated students in particular, given the consistently high rates of sexual assault occurring in this high-risk group.

### 1.1. Rape Myths

Burt [2] defined rape myths as “prejudicial, stereotyped or false beliefs about rape, rape victims and rapists, in creating a climate hostile to rape victims” (p. 217). Burt created the first validated measure to assess people’s endorsement of these rape myths still in use to some extent today. Interestingly, all of the items focus primarily on victim-related myths (e.g., women who drink or are promiscuous are at fault for rape). Lonsway and Fitzgerald [11] provided a thorough review of approximately 24 studies that assessed rape myths, but few studies addressed stereotypes regarding perpetrators of rape. Payne, Lonsway and Fitzgerald [12] developed the Illinois Rape Myth Acceptance (IRMA) scale, which expanded Burt’s original measure; however, it still remained focused on myths related to rape victims or misunderstandings related to the nonconsensual encounter or women’s behavior thought to contribute to rape. Lonsway and Fitzgerald [11] concluded that the literature was far more focused on victim-blaming, rather than focusing on stereotypes of perpetrators. Both Payne and colleagues’ [12] and Burt’s [2] measures reflect this notion as both remain focused on myths related to victims and sexual assault experiences, with little attention focused on perpetrators’ rape myths.

For decades, researchers have found that rapists are typically stereotyped as being poor and as belonging to a racial/ethnic minority group [13]. As such, poor men of color are stereotyped as “bad guys”, while white, middleclass/affluent and educated men are seen as “good guys”. Lisak [14] argues that most college-aged men believe that a typical rapist wears a ski mask, carries a knife and attacks strangers in dark corners. He argues this false stereotype exists because most college men do not fit that profile; that is, most college men do not attack strangers in dark corners with knives. As such, middle-class, white college men do not label themselves or others like them as rapists. In other words, when these men see a “good guy” (a middle-class, college student accused of sexual assault), they are less inclined to label him as a rapist; we have called this conceptualization the “good guy stereotype”. We theorize that there are rape myths surrounding who commits rape; however, preconceived stereotypes dictate that rapists are not middle class “good guys”. The present study acknowledges the widely-established field of rape myths, but provides an additional aspect of rape myth acceptance by focusing on the stereotypes and myths regarding perpetrators of rape.

### 1.2. Greek Life

On college campuses in the U.S., Greek life is both prevalent and is often comprised of homogenous students; they are generally more affluent and are more likely to be white [15,16]. This often translates into having more power and privilege on campus and being part of an elite social class [15,17] because they are from more affluent families, have higher socioeconomic status and have larger social networks [18,19]. Second, Greek-affiliated student life is heavily centered on heteronormative behavior, party culture and popularity in the form of social hierarchy [15,16,17,20]. Those who participate in Greek-life, especially fraternity men, endorse more traditional gender roles, engage in more sexually aggressive behaviors, are more accepting of rape myths and more strongly endorse hostility toward women; fraternity men also consume larger amounts of alcohol and drugs and place a higher value on social life including partying and socializing with peers [7,21,22,23,24]. In addition, there is increased comfort between members in Greek-life, due to the Greek relationship of family [16,25]. Given that sorority women frequently interact with fraternity men, sorority women may not perceive their risk of victimization at a high level if surrounded by brothers and sisters, compared to other university groups [25].

Recently in 2015, men from the Pennsylvania State University’s chapter of Kappa Delta Rho were being investigated for an alleged secret Facebook page (accessed only via invite) called Covert Business Transactions. Pictures of naked, semi-naked and potentially unconscious women were posted on this page without the permission of at least some, if not most of the women. This page was allegedly secretly shared with approximately 144 current and former members of the fraternity. According to Penn State’s President Eric Barron, the Facebook posts were “very sad and very offensive …. It’s just, unfortunately, a large system with some very fine young men and some men who are not doing smart things” [26]. On the one hand, one could argue that some of Kappa Delta Rho’s members were perhaps unaware of the Facebook page. On the other, if such a large number of current and former chapter members were participating in this page and saw no problem in taking pictures of women without their permission, it stands to reason that there is a systematic problem. This problem is potentially facilitated by the mentality that Kappa Delta Rho are “fine young men … who are not doing smart things,” rather than deviant men, behaving in malicious ways in this context.

Jozkowski [27] discusses the idea that some men can be “good” in certain contexts, but potentially dangerous and assaultive in other contexts. Accordingly, she states “This (the fact that seemingly “good” men perpetrate sexual assault) is something that we grapple and need to come to terms with as a culture: Men can be “good” in certain contexts, yet simultaneously perpetrators of sexual assault.” This disconnect may be due to perceptions of who perpetuates sexual assault. As such, good individuals would ultimately not fit the profile of someone who rapes; essentially, good guys do not rape. Because fraternity men seem to embody such good qualities (e.g., higher GPA and more likely to graduate than non-fraternity students, as well as greater philanthropy efforts; [28]), they are often conceptualized as “good guys”. Although fraternity men may be perceived as inherently “good”, fraternity men are more likely to perpetrate sexual assault on college campuses [6,8,10]. In fact, according to Armstrong, Hamilton and Sweeney [20], fraternity men construct a dynamic at their parties and social events that actually facilitates what they call “low-level sexual coercion”, which in turn contributes to what they call a “party rape culture”. Yet, universities may be reluctant to chastise fraternities for such behavior. Reasons for avoiding ramifications against fraternities may be financially driven (e.g., Greek alumni are the largest donors to universities and offset costs of university housing by funding their houses; [18]) or based on the belief that fraternity men are primarily “good boys” with only “a few bad apples” to blame for such bad behavior [17,29]. Thus, it is important to examine how people view perpetrators of sexual assault. In particular, it may be exceedingly important to assess these attitudes among individuals involved in Greek-life (both fraternity men and sorority women), given their high risk for sexual assault perpetration and victimization.

### 1.3. The Current Study

The current study aims to examine how college students, and in particular fraternity men and sorority women, view perpetrator rape myths. In line with previous research examining rape myths [1,3,4,5,23], we hypothesized that Greek students would favor more stereotypical perpetrator rape myths as compared to non-affiliated Greek students. Specifically, we also hypothesized that Greek-affiliated men would endorse more stereotypical perpetrator rape myths than sorority women and unaffiliated men and that sorority women would endorse more stereotypical perpetrator rape myths compared to unaffiliated women.

## 2. Methods

### 2.1. Procedure/Participants

Data were obtained from a convenience sample of college students at a large public university located in the southern United States. Students who were at least 18 years of age and enrolled in classes at the university were recruited primarily in general education and elective health and family sciences undergraduate courses, as well as a university-wide newsletter. These courses were selected because they are taken by a range of students with regard to year in school and course major. Participants completed an anonymous online survey via Qualtrics. Students could enter their name into a drawing for a $50 gift card by supplying their email; when permitted by the course instructor, some students received extra credit in their courses as an incentive for survey completion. IRB approval was granted from the university for data collection.

There were 942 students who completed the survey. Because we were interested in traditional college-aged young adults (18–25), we excluded 42 individuals because they were over the age of 25. We also removed 8 individuals who identified as transgender as the sample size was too low to make meaningful comparisons. The final sample consisted of 892 college students; 22% male (*n* = 193) and 78% female (*n* = 699), with a mean age of 20.07 (*SD* = 1.36). The sample was primarily Caucasian (*n* = 707; 79%), with 6% identifying as black/African American, 6% Hispanic/Latino and 9% identifying as other. Ninety-six percent of respondents identified as heterosexual (*n* = 859). Over half of the participants (58%, *n =* 513) were currently or had previously been Greek-affiliated. Second- (32%) and third-year students (32%) comprised the largest class standing categories, followed by fourth-year students (21%), first-year students (13%) and other (i.e., graduate, non-degree students, 2%). Most respondents were single, not actively dating (37%) or in a committed relationship (35%).

### 2.2. Measures

Controls: Previous victimization was hypothesized to influence participants’ perceptions regarding sexual assault, making it a necessary control variable. We assessed sexual assault victimization via a modified version of the Modified Sexual Experiences Survey-Short Form Victimization (SES-R). The SES-R determines an individual’s nonconsensual sexual experiences from those that occurred before and since the age of fourteen; the current study assessed victimization since the age of 14. The scale measures seven types of unwanted sexual experiences (including unwanted sexual touching, oral, vaginal and anal penetration) by five different methods the perpetrator may have used (i.e., verbal coercion, manipulation, taking advantage via alcohol/drugs, physical threatening and physical force) and aims to identify previous victimization while avoiding terms such as rape, due to vastly varied definitions from respondents [30]. Participants were coded into two categories based on their responses, victim and non-victim. There were 114 participants who reported rape victimization as experiencing completed nonconsensual oral, anal or vaginal penetration (13%; 109 women, 5 men). A majority of the victims were affiliated with Greek-life (*n* = 73; 64%).

Rape myth acceptance was measured by the Illinois Rape Myth Acceptance Scale, which assesses a participants’ support for rape myths or false beliefs about sexual assault and sexual assault victims [12]. Example items include: “Rape happens when a man’s sex drive gets out of control,” “Many women secretly desire to be raped,” “If a guy is drunk, he might rape someone unintentionally” and “A rape probably didn’t happen if the girl has no bruises or marks.” The scale ranged from 1 = strongly disagree to 7 = strongly agree (*M* = 2.36, *SD* = 1.03, alpha = 0.94) and was used in order to assess the convergent validity of the perpetrator rape myth measure.

Perpetrator rape myths were measured by items specifically written for the current study. These items were written to understand ways in which people view perpetrators better, in terms of stereotypical rape scenarios. Guided by previous research [12,31,32,33], 20 items were written to assess the characteristics and stereotypes of those who perpetrate sexual assault; see Table 1 for all items. The 20-item measure ranged from 1 = strongly disagree to 7 = strongly agree *(M* = 2.08, *SD=* 1.00, alpha = 0.94).

In order to assess the convergent validity of perpetrator rape myths, we examined the association with rape myth acceptance. Perpetrator rape myths were positively and strongly correlated with the rape myth acceptance scale (*r* = 0.64, *p* < 0.001). Students who reported higher rape myth attitudes also perceived perpetrators as more stereotypical.

Analyses: To examine the perpetrator rape myth measure further, principal axis factoring (PAF) was used to assess the items. Initially, eigenvalues and the scree plot were utilized to determine the number of factor loadings; factors with an eigenvalue > 1 were considered to be significant [34] and were thus retained. We also used a combination of theory and statistical results post item elimination [35] to determine which items to retain. In order for an item to be retained, a factor-loading cutoff was established at 0.5 [36,37]. This resulted in one factor. It was determined that 7 items (e.g., (1) sexual assault victims often personally know their rapist; (2) women are more likely to be raped by men that are the same race as them; (3) white people are more likely to rape than racial/ethnic minorities; (4) rape rarely happens in the victim’s own home/dorm/apartment; (5) rape does not happen at a party with other friends around; (6) men who rape only rape strangers; (7) rape happens on the bad side of town) did not load at 0.5 or higher on any factor, cross-loaded at 0.5 on multiple factors or did not assess theoretically “good guy” stereotypes, focusing more on location or relationship between the victim and perpetrator. The remaining 13 items in the perpetrator rape myth measure resulted in an alpha of 0.92 (see Table 1) and included items that measured specifically good guy items, such as attractiveness, high socioeconomic status and likeability. We also assessed specific items that referred to athletes and fraternity men. Items in reference to the bad guy factor were based on stereotypical perpetrator depictions, such as non-white ethnicity and lower socioeconomic status. The items theoretically matched the concepts based on those who are seen as good guys do not rape and those who are seen as bad guys do rape. Although we expected two factors to emerge, only one overall 12-item factor emerged. Lastly, an ANCOVA was used to test the hypotheses examining gender and Greek status on perpetrator rape myths, while controlling for previous victimization and rape myth attitudes.

## 3. Results

Using an ANCOVA, findings indicated, above and beyond previous victimization and rape myths, a significant effect for gender (*F* = 49.94, *p* < 0.001) and for Greek status (*F* = 11.57, *p* < 0.001). Men reported more stereotypical perpetrator rape myths (*M* = 2.52) as compared to women (*M* = 2.08), and Greek-affiliated students reported more stereotypical perpetrator rape myths (*M* = 2.41) compared to non-affiliated Greek students (*M =* 2.20), thus supporting our first hypothesis. Lastly, there was a significant gender by Greek interaction (*F* = 4.98, *p* < 0.05). To follow up on the interaction, four groups were created based on gender and Greek status: sorority women, non-affiliated women, fraternity men and non-affiliated men (see Table 2). Contrary to our second hypothesis, there was no significant difference between sorority women and non-affiliated women on perpetrator rape myths. However, consistent with our second hypothesis, there were significant differences among fraternity men and all other subgroups (i.e., non-affiliated men, sorority women and non-affiliated women) and between non-affiliated men and both groups of women (i.e., sorority women, non-affiliated women), indicating that both groups of men reported more stereotypical perpetrator rape myths compared to both groups of women. Fraternity men reported the highest stereotypical perpetrator rape myths compared to all groups, including non-affiliated men. Notably, fraternity men did not strongly agree with perpetrator rape myths (which would be indicated by an average mean of seven); however, their answers were significantly higher than all other group responses. Because only fraternity men reported significantly higher responses compared to non-affiliated men and there was no difference between sorority and non-affiliated women, our second hypothesis was partially supported.

## 4. Discussion

The goal of the current study was to better understand college students’ perceptions of perpetrator rape myths. Students who held stereotypical perceptions of those who commit sexual violence also more strongly endorsed negative attitudes about rape victims. Previous measures of rape myths focus mainly on stereotypical views of women as victims of sexual assault [12], while our proposed assessment of perpetrator rape myths, similarly, focused on stereotypical views of perpetrators, who are primarily men in cases of sexual assault [38]. Conceptually, the association of these measures makes sense as they both assess stereotypical views of those involved in sexual violence: negative attitudes about women who experience rape (e.g., women are asking for it; [12]), as well as stereotypical views that good guys (e.g., attractive, high GPA, active in student groups) do not rape.

The current study found that, in general, men held stronger stereotypes regarding perpetrators of rape compared to women. Previous literature has shown that men more highly endorse rape myth acceptance compared to women [11,31]. Thus, our findings regarding perpetrator rape myths are fairly consistent as both female subgroups were significantly different from the male subgroups.

The study also found that college students in Greek-life, specifically fraternity men, held more stereotypical perceptions regarding perpetrators of sexual assault. This is supported by previous research in which men involved in Greek-life endorse more traditional gender roles and higher rape myth acceptance [1]. Previous research has not examined the perceptions of fraternity men on who they view as sexual assault perpetrators. As predicted, Greek-affiliated men held stronger beliefs that “good guys don’t rape”. Greek men, unlike other college students, are the ideal in-group on the college campus. Fraternity men control party resources [20], are positioned at the top of the social hierarchy [16,17] and have high levels of group loyalty and secrecy [10,16]. Thus, perhaps it is not entirely surprising that Greek men would hold more favorable views about “good guys” and not perceive other good guys (or themselves) as perpetrators of sexual assault. Greek men also held stronger views as compared to other men, who are non-affiliated with Greek-life. However, it is important to note that they did not strongly agree with perpetrator rape myths, but that their answers were significantly higher than all other group responses. Thus, our finding should be approached with caution. Additional research is needed to assess this scale development of perpetrator rape myths. Future research could also utilize qualitative interviews to determine why fraternity men were less likely to disagree with these responses compared to other students.

Contrary to what was hypothesized, there was no difference between Greek-affiliated women and non-affiliated women. Sorority women are more likely to interact with fraternity men [25] and more likely to be victims of sexual assault [7,39,40]. In fact, the current study found that 64% of sexual assault victims were affiliated with Greek-life. Perhaps sorority women are becoming more aware of their potential risk for sexual assault than previous research has demonstrated. For example, in a qualitative study conducted by Norris and colleagues [25], sorority women showed a relatively high degree of awareness for their general risk regarding sexual aggression, as well as possible prevention mechanisms to help other women (i.e., watching out for other women who drank too much, buddy system, hand signals used to signal for help). Future research should examine if sorority women perceive perpetrators of sexual assault more accurately because of their increased exposure to “good guys” (i.e., fraternity men).

### 4.1. Strengths and Limitations

The current sample was primarily Caucasian women, thus the greatest limitation to this study was the lack of male participation. Having a larger amount of those affiliated with Greek-life would be beneficial. Notably, the sample of fraternity men in this sample was relatively small, and there were also small cell sizes for the ANCOVA; this increases vulnerability to error. The sample was collected from a campus with a high Greek-life student body located in the southern U.S., meaning that findings should not be considered generalizable to other types of universities and college campuses (e.g., satellite campuses, community colleges, liberal arts schools). This research was focused on the typical social Greek-life, where participants are most likely to be Caucasian and upper middle class [17]. However, future research should expand by using a more diverse sample to further examine the differences with fraternities and sororities across the U.S., including small and large campuses in different geographic locations. Lastly, the current study attempted to “pilot” test perpetrator rape myths; however, a few items appeared to measure other rape myths, such as environmental factors (e.g., “bad side of town”). Additional research is needed to test our proposed construct.

### 4.2. Implications

Public perceptions and expectations of rape and the context in which it takes place are different from reality [31]. Oftentimes, the media focuses on sexual assault cases that do not reflect the norm (i.e., which then upholds rape myths) and instead focuses on cases where the perpetrator fits a typology of a creepy man in the bushes [2,27,41]. Research has shown the framing of sexual assault through the media directly affects attitudes about rape [42,43,44]. For example, stranger rape and false accusations that are given large amounts of media attention are seen as more legitimate and offer a frame of reference when determining the legitimacy of other cases [41,45]. In addition, media typically focuses on the “myth of the black rapist,” when the story is of a black perpetrator and a white women [46]. This allows for the public to recall these situations or schema and to believe they are more common than they truly are [31,47]. Instead, the media needs to reflect that perpetrators of sexual assault can look like the “boy next door” and not the creepy man lurking in the bushes, who rarely exists [27]. Thus, it is difficult to hold men who perpetrate sexual violence accountable if guilt is difficult to attribute to men who have good characteristics in other aspects of their lives, outside of sexual violence. Typically, when college-aged men are accused of rape, society perceives that they were essentially “good guys who drank too much alcohol and who used poor communication” [14]. Criminal justice research suggests that perceptions of perpetrators play a significant role in peoples’ beliefs about the “legitimacy” of sexual assault as a crime and the manner in which guilt is determined [41,42,45,48]. Perceptions of sexual assault often develop from rape scripts, which endorse a stereotypical view of rape experiences [49]. That is, rape scripts present “real rape” as a violent attack perpetrated by a stranger in a public location [50,51]. Despite the endorsement of these scripts, the majority of sexual assaults qualify as acquaintance rapes, occurring between two people who know each other (i.e., up to 90%; [52]). Thus, rape myths and rape scripts are mutually reinforcing, defining rape more narrowly, and inaccurately. Unfortunately, endorsement of these rape myths and false rape scripts often results in increased victim blaming and shielding perpetrators from blame [49].

In a recent and controversial case in 2016, a Stanford athlete named Brock Turner was found guilty of raping a young unconscious female. Although prosecutors argued for at least a six-year sentence, Judge Aaron Persky sentenced the rapist to only six months of prison because his rationale was that Brock Turner’s “… positive character references given by his father had factored into his decision, as well as his age, his lack of a criminal history, and the role that alcohol played in the assault …. A prison sentence would have (too) a severe impact on him” [53]. In addition, the rapist fit the same profile as the judge: White, middle-class, Stanford athlete alumni and a Phi Beta Kappa. Thus, the rapist was a “good guy.” However, voters in 2018 did not agree with Judge Aaron Persky’s ruling, as he was the first judge recalled in California in more than 80 years [54]. The incident at Stanford University is certainly not in isolation as there are a myriad of sexual assault cases making national news as of late that involve men whom we would otherwise conceptualize as “good guys.” Educational efforts aimed at shifting public view of what a rapist looks like, which may include those who are deemed “good guys” is clearly warranted.

Although the Interfraternity Council (IFC) that oversees Greek-life does not publish statistics on Greeks, we do know that fraternity and sorority alumni represent the “largest sector of lifetime donors to colleges, four times more than non-Greeks, and thus have a firm grip on university politics” [18]. Thus, it seems that fraternity men hold a lot of power on college campuses and are largely represented as perpetrators of sexual assault [4,6], and yet, few people think that they, the good guys, could ever be potential rapists. As a society, we need to increase awareness of rape culture on campuses, as well as the perceptions of how rape happens on college campuses.

In conclusion, previous research has focused on perceptions of victims (i.e., rape myth acceptance), and to our knowledge, this is the first study that has focused specifically on assessing perpetrator rape myths. Future research should continue to develop better measures to assess rape myths that include stereotypes about perpetrators. Future research is warranted to better understand how Greek-affiliated alumni holding high power offices (prosecutors, judges, Title IX Coordinators) may perceive perpetrators of sexual assault in their decision-making. Research has shown the framing of sexual assault as more stereotypical through the media directly affects attitudes about rape [42,43,44], and so, it is possible that this would spill over into the criminal justice system and Title IX decisions on college campuses.

## Figures and Tables

**Table 1 behavsci-08-00060-t001:** Perpetrator rape myth items.

	*M*	*SD*
**Perpetrator Rape Myth Items (*a* = 0.94)**	2.17	1.00
Good guys do not rape.	2.63	1.77
Men with high GPAs do not rape.	1.88	1.16
Men with a lot of friends will rape.	2.82	1.64
Men who are actively involved in student clubs do not rape.	2.03	1.23
Good looking guys do not rape.	1.80	1.11
Guys who are well liked by others will not rape.	1.87	1.16
Men from good families do not rape.	2.00	1.29
A rapist is more likely to be black or Hispanic than white.	2.23	1.39
Men who are in lower socioeconomic status or social class are more likely to rape.	2.57	1.54
Men from nice middleclass homes almost never rape.	1.93	1.18
College athletes are less likely to rape because women always want to have sex with them.	1.82	1.19
Women are always looking to have sex with college athletes, so there is no need for them to rape.	1.83	1.22
Fraternity men often get accused of rape when women regret consensual sex.	2.69	1.61

**Table 2 behavsci-08-00060-t002:** Perpetrator rape myths as a function of gender and Greek affiliation.

	Gender and Greek Affiliation					
	1 Sorority Women *n* = 440	2 Non-Affiliated Women *n* = 281	3 Fraternity Men *n* = 73	4 Non-Affiliated Men *n* = 138	*F*	*ή* ^2^
Perpetrator Rape Myths	2.11 ^ab^	2.03 ^cd^	2.71 ^ace^	2.33 ^bde^	17.12 *	0.06

Note: Matching letters indicate significant differences using Bonferroni adjustment. Analyses control for previous victimization (*M* = 0.13) and rape myths (*M* = 2.35). * *p* < 0.001.

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
