# Peer review of "“Good Guys Don’t Rape”: Greek and Non-Greek College Student Perpetrator Rape Myths"

_behavsci, 2018, doi:10.3390/bs8070060_

Round 1

Author Response

Reviewer 1: Overall, I think the manuscript has some strengths in that it draws attention to rape myths among college students and attempts to delve deeper into category of “perpetrator rape myths.” However, I have serious concerns that led me to recommend the manuscript be rejected, perhaps with the possibility to resubmit if major revisions are made. My primary concerns are (1) lack of construct explication in the literature review, particularly around the idea of perpetrator rape myths, (2) insufficient discussion of the hypothesized group differences by gender and Greek-life participation, and (3) a fatal flaw in data analysis when they incorrectly used PCA to analyze the perpetrator myth items. In terms of writing quality, I did not identify any major problems. There were a few sentences with awkward construction. For example, line 62 the authors mention “good qualities.” Good qualities according to whom?

Thank you for reviewing our paper and giving us another opportunity to address these major issues. We agree with your assessment and have updated the paper to reflect your suggestions. First, we fixed issues in our sentence construction. We have responded to your items below.

Introduction

§  The introduction was somewhat difficult to follow, with only shallow discussion of the rape myths construct. Why do we need a separate measure of perpetrator rape myths? To some extent this is already captured by IRMA (“he didn’t mean it” subscale). You essentially rolled a scale development project into this manuscript but with very little in the literature review to back it up.

We agree that we were missing stronger justification for our developed measure, and have set this up better in the literature review section, especially in how our measure relates to previous rape myths scales.

§  I appreciated the authors’ attempt to describe the context of Greek life, but the rationale for the group level comparisons (gender and Greek-life affiliation) could have been better explained. If you think that privilege on campus (racial, socioeconomic, social) plays a role, I strongly suggest you measure these variables. I suspect Greek affiliation may be functioning as a proxy for these other privileged social positions that are associated with greater rape myth agreement

We agree and have attempted to justify our reasoning for examining Greek Life students.

§  You only briefly discussed women’s experiences (lines 53-55). What is your rationale for the hypothesized difference between men and women’s perpetrator rape myth acceptance.

We have included this change by including literature that supports a gender effect, as well as a Greek effect on differences in rape myths.

§  Seemed to be missing some citations (e.g., line 72 – who is making this argument about “bad apples”? you?); Line 80 (date for quote from Penn State president)

We have referenced these quotes.

 Methods

§  Which courses were students recruited from?  Primarily health and family science courses, which are also primarily general education or general elective courses taken by a range of students in different course majors (now included)

§  Typo on line 144 of -> for  - This sentence was rewritten.

§  Lines 145-146, you should say “traditional aged college-students” – by virtue of being in college, your older participants were college-aged – We have added “traditional”

§  You briefly mention this in the discussion, but it’s a serious concern that your cell sizes for the ANCOVA are so uneven. This increases vulnerability to error. We have included this in our limitations section as it is an important issue that we failed to acknowledge.

§  Did you only ask about White vs. “non-White”? This is a reductive way to present racial/ethnic identity. If you have any further information about the “non-White” group, you should present it. We have included this information about our sample.

§  I agree that controlling for previous victimization makes sense. What was your rationale for categorizing victim/non-victim based only on completed oral, anal, or vaginal penetration?  Attempted sexual assault could conceivably have a meaningful impact on respondent’s rape myth agreement.

We agree, and the numbers of those who have been assaulted would be much higher, but we felt that using completed perpetration was a more conservative way to measure previous assault, and it was notable that 13% of our sample indicated completed rape (much higher than previous research).

§  Need to present reliability and validity evidence for the IRMA We have included this in the measures.

§  Comments specifically related to measurement of “perpetrator rape myths”

·      Problems with measurement of perpetrator rape myth agreement is my biggest criticism of this manuscript. It is inappropriate to use PCA in scale development (e.g., Worthington & Whittaker, 2006). This has been well documented and is an egregious error. It calls into question all subsequent analyses. If you decide to revise this manuscript, you must go back to this point and run PAF instead. Thank you for letting us know about this error, and we have now run the analyses using PAF instead and have changed the results accordingly.

·      Having only one a priori retention criterion (eigenvalue > 1) is insufficient. Review scale development guidelines for suggestions. You say this was designed to measure perpetrator rape myths but a few items appear to tap into other myths re: environmental factors (e.g., “bad side of town” item in the final version). This increases risk of construct conflation. Where did these items come from, who authored them, did you do a pilot test of items with college students?  The items come from various scales that assessed “rapists”: Guided by previous research (Edwards, Turchik, Dardis, Reynolds, & Gidycz, 2011; Lievore, 2004; McKimmie et al., 2014; Payne et al., 1999), 20 items were written to assess characteristics and stereotypes of those who perpetrate sexual assault.

Analyses and Discussion:

§  I have provided limited feedback on this section, do to my concerns about “up line” problems. Your means are so, so low. This is a good thing in that not many of your respondents were agreeing with the perpetrator rape myths. Although the differences may have reached the level of statistical significance, the actual difference between groups is not very compelling from an applied/clinical standpoint. This is reflected in your very low effect size for group on perpetrator rape myth (.06). In line 230, you say that Greek men did not strongly agree – they weren’t even close. On average, Greek men disagreed with the perpetrator rape myths. This affects the implications of your findings.

We agree that we need to make sure our readers know that the groups were quite similar, significantly different, but not likely to be that meaningful given the similar rates. We have reflected this in our discussion and the need for future research to continue addressing and measuring perpetrator rape myths.

Reviewer 2 Report

Overall, the study and the article are good.  I would recommend a close reading/proofreading of the Introduction and Greek Life sections for grammar, sentence structure, and accurate word choice (e.g., "dangerous" for "dangers" in line 87.  Some words are hyphenated which shouldn't be.

I think the Introduction needs a little more background and information included.

I was initially concerned about the lack of male participants for the study and had questions about the ability to draw meaningful conclusions.  I think the authors handled this well in the Limitations section and I was pleased this was addressed.

I believe there could be additional discussion added about the finding of the % of women in the sample who have been assaulted and are now affiliated with Greek life.  While it isn't clear whether the assaults occurred prior to or during this affiliation, it is an interesting point of discussion and could be added to the future research section as well.

Author Response

We appreciate the reviewers’ comments on how to make this paper more acceptable for publication. We have made changes that the reviewers requested. We highlighted in green all the changes in the manuscript, as well as provided responses to each reviewer comment.

Reviewer 2: Overall, the study and the article are good.  I would recommend a close reading/proofreading of the Introduction and Greek Life sections for grammar, sentence structure, and accurate word choice (e.g., "dangerous" for "dangers" in line 87.  Some words are hyphenated which shouldn't be.) I was initially concerned about the lack of male participants for the study and had questions about the ability to draw meaningful conclusions.  I think the authors handled this well in the Limitations section and I was pleased this was addressed.

Thank you for reviewing our article and providing positive feedback, as well as areas where we can improve. We fixed “dangerous” and deleted many hyphenated words that were not necessary.

I think the Introduction needs a little more background and information included.

We have attempted to tighten the introduction by reorganizing, and also provided more literature to support our study hypotheses and goals, especially concerning the association between rape myths and our perpetrator rape myth scale.

I believe there could be additional discussion added about the finding of the % of women in the sample who have been assaulted and are now affiliated with Greek life.  While it isn't clear whether the assaults occurred prior to or during this affiliation, it is an interesting point of discussion and could be added to the future research section as well.

This is an excellent point--we included a small discussion on this finding.